# Comparison of Six Antifungal Susceptibilities of 11 *Candida* Species Using the VITEK2 AST–YS08 Card and Broth Microdilution Method

Hyeyoung Lee,[a] Seong Hyouk Choi,[b] Junsang Oh,[c] Jehyun Koo,[a] Hyun Ji Lee,[a] Sung–Il Cho,[a] Jeong Hwan Shin,[d] Hae Kyung Lee,[e] Soo–Young Kim,[f] Chae Hoon Lee,[g] Young Ree Kim,[h] Yong–Hak Sohn,[i] Woo Jin Kim,[j] Sook Won Ryu,[k] Gi–Ho Sung,[c,l] Jayoung Kim[a,c]

aDepartment of Laboratory Medicine, International St. Mary's Hospital, College of Medicine, Catholic Kwandong University, Incheon, Korea

bDepartment of Hematopoietic Stem Cell Transplantation Center, Soonchunhyang University Bucheon Hospital, Bucheon, Korea

cTranslational Research Division, Biomedical Institute of Mycological Resource, International St. Mary's Hospital, College of Medicine, Catholic Kwandong University, Incheon, Korea

dDepartment of Laboratory Medicine, Inje University College of Medicine, Busan, Korea

eDepartment of Laboratory Medicine, College of Medicine, The Catholic University of Korea, Uijeongbu St. Mary's Hospital, Uijeongbu, Korea

fDepartment of Laboratory Medicine, College of Medicine, The Catholic University of Korea, St. Vincent's Hospital, Suwon, Korea

gDepartment of Laboratory Medicine, Yeungnam University College of Medicine, Daegu, South Korea

hDepartment of Laboratory Medicine, Jeju National University College of Medicine, Jeju, Korea

iDepartment of Laboratory Medicine, Seegene Medical Foundation, Seoul, Korea

jDepartment of Laboratory Medicine, EONE Laboratories, Incheon, Korea

kDepartment of Laboratory Medicine, Kangwon National University School of Medicine, Chuncheon, Gangwon–do, Korea

lDepartment of Microbiology, College of Medicine, Catholic Kwandong University, Gangneung, Korea

**ABSTRACT** We used a Vitek 2 AST–YS08 (YS08) system and the broth microdilution method (BMD) adopted by the Clinical and Laboratory Standards Institute (CLSI) to compare the susceptibility of 184 isolates of 11 *Candida* species to fluconazole, voriconazole, micafungin, caspofungin, amphotericin B, and flucytosine. In *Candida albicans*, the categorical agreement (CA) was 79.2%, 91.7%, 95.8%, and 95.8% for fluconazole, voriconazole, micafungin, and caspofungin, respectively. About 12.5% and 4.2% of very major errors were detected for fluconazole and voriconazole, respectively. *C. glabrata* showed excellent essential agreements (EAs) (>90%) for azoles but different MIC distributions for fluconazole and caspofungin. The CA between BMD fluconazole MICs and YS08 voriconazole MICs by the method-specific clinical breakpoint (CBP) was 90% in *C. glabrata*. Over 80% of *C. glabrata* and *C. krusei* isolates identified as micafungin–susceptible were labeled intermediate or resistant to caspofungin in YS08. In *C. parapsilosis*, 5.3% of very major errors and 10.5% of minor errors were found, whereas 33.3% of minor errors were observed in *C. tropicalis* for fluconazole. For *C. tropicalis*, 13 (61.9%) non-wild type (WT) isolates of fluconazole and 7 (33.3%) non-WTs of voriconazole were classified in YS08 as WT. For *C. auris*, the EAs were 93.3%, 100%, 82.2%, 97.8%, and 97.8% for fluconazole, voriconazole, micafungin, caspofungin, and amphotericin B, respectively. YS08 showed comparable results to the BMD. However, considering the lower YS08 fluconazole MIC results compared with BMD in *Candida* species and YS08 caspofungin results in *C. glabrata* and *C. krusei*, improvements are needed.

**IMPORTANCE** The new Vitek 2 AST–YS08 (YS08) card has been updated to reflect the recently revised Clinical and Laboratory Standards Institute (CLSI) guideline. In this study, antifungal drug susceptibility tests were performed using the YS08 card and compared with the CLSI broth microdilution (BMD) method. In conclusion, YS08 showed similar results to BMD, including with *C. auris*. However, about 12.5% and 4.2% of major errors were detected for fluconazole and voriconazole, respectively, in

Address correspondence to Jayoung Kim, Lmkjy7@gmail.com, or Gi–Ho Sung, sung97330@gmail.com.

The authors declare no conflict of interest.

*C. albicans*. More than 80% of *C. glabrata* and *C. krusei* isolates identified as susceptible to micafungin were labeled moderate or resistant to caspofungin in YS08. The categorical agreement between BMD fluconazole MICs and YS08 voriconazole MICs was 90% by the method-specific CBP of voriconazole, 80% by the current epidemiological cutoff value (ECV) (0.25 $\mu$g/mL) of voriconazole, and 85% by the previous ECV (0.5 $\mu$g/mL) of voriconazole. Further improvements in YS08 for the detection of fluconazole and echinocandin resistance are thus needed.

**KEYWORDS** *Candida*, antifungal susceptibility, Vitek 2 AST–YS08, broth microdilution, epidemiological cutoff value

Candida species are normal commensals that localize on the skin and mucosal membranes of genitals and the gastrointestinal tract. However, they can cause various infections in vulnerable patients, such as the elderly, hospitalized, or immunosuppressed (1). Invasive fungal infection due to *Candida* species is widely recognized as a leading cause of morbidity and mortality in medical environments (2). The mortality rate from invasive candidiasis is estimated to be around 19–40% (3), and is much higher in intensive care unit patients, approaching 70% (4). The distribution of species has changed over the past decades (3). *Candida albicans* was previously the predominant pathogen (3, 5), but in recent years, *C. glabrata*, *C. tropicalis*, *C. krusei*, *C. parapsilosis*, and *C. lusitaniae* have emerged as important pathogens (6). According to epidemiologic surveillance data from Korea, *C. tropicalis* (36.4%) is the most common non-albicans *Candida*, followed by *C. glabrata* (28.5%), *C. parapsilosis* (24.7%), and *C. krusei* (2.6%) (7). *Cyberlindnera fabianii* (*Candida fabianii*) is an uncommon opportunistic yeast species, but it has the ability to cause septicemia and rapidly acquire resistance to fluconazole and voriconazole (8).

Antifungal agents target various biosynthetic pathways of pathogens. Echinocandins target biosynthesis of the cell wall. Azoles target the important enzyme 14$\alpha$–demethylase in ergosterol biosynthesis. Flucytosine (5–FC) interferes with nucleic acid biosynthesis. Polyene drugs, including amphotericin B, bind with ergosterol to form pores and are fungicidal (1, 9).

An important factor that may contribute to therapeutic failure is antifungal agent resistance (1). Increased MICs and other specific mechanisms of resistance are associated with treatment failure and mortality, making empirical treatment choices difficult for clinicians (10). The molecular mechanisms involved in antifungal resistance include overexpression of membrane transporters, changes in the biosynthesis of the cell wall and ergosterol, mutations in the transcription factors that regulate membrane transporters, and ergosterol biosynthesis (9).

In addition, the emergence of multidrug-resistant *C. glabrata* and *C. auris*, the increase of fluconazole-resistant *C. tropicalis* and *C. parapsilosis*, and the existence of the intrinsically resistant *C. krusei* are more problematic (11). *C. auris* has emerged as a new multidrug-resistant species that causes a wide range of infections, especially in intensive care units (12). In 2016, the Infectious Diseases Society of America recommended antifungal susceptibility testing for azoles in clinically relevant *Candida* isolates and all bloodstream infections (2).

Broth microdilution (BMD) is considered the most reliable reference method for the evaluation of antifungal susceptibility in *Candida* species (13). The Clinical and Laboratory Standards Institute (CLSI) has developed a standard method and established clinical breakpoints (CBPs) for the most common *Candida* species (14). The epidemiological cutoff value (ECV) identifies isolates that have a non-wild type (WT) profile, and CLSI has updated its documentation to provide ECVs for the less prevalent *Candida* species (15).

Vitek 2 (bioMérieux, Marcy l'Etoile, France) is a fully automated system capable of performing microbial identification and susceptibility testing. The system can be applied to bacteria and yeast and is used by many laboratories because of its rapidity and ease of use (16). The new Vitek 2 antimicrobial susceptibility testing (AST) system

for yeast, AST-YS08 card, has been updated to reflect the recently revised CLSI CBP for common *Candida* species (17).

The aim of this study was to evaluate the clinical applicability of the new Vitek 2 AST–YS08 (YS08) card by comparing it with the results of the BMD method by CLSI.

## RESULTS

MIC distributions for each *Candida* species were determined by CLSI reference broth microdilution method (Table 1). For *C. albicans*, the categorical agreement (CA) was 79.2%, 91.7%, 95.8%, and 95.8% for fluconazole, voriconazole, micafungin, and caspofungin, respectively. The very major errors of 12.5% (3/24) for fluconazole and 4.2% (1/24) for voriconazole were detected in *C. albicans*. The essential agreements (EAs) between the BMD and YS08 were 83.3%, 91.7%, 95.8%, 95.8%, and 95.8% for fluconazole, voriconazole, micafungin, caspofungin, and amphotericin B, respectively, in *C. albicans*.

In *C. glabrata*, the CA was 100% for micafungin but only 15% for caspofungin between the BMD and YS08. A major error of 10% (2/20) and minor error of 75% (15/20) for caspofungin were detected. In YS08, 85% (17/20) of isolates were interpreted as caspofungin-intermediate (15 isolates) or -resistant (two isolates), but all showed micafungin susceptibility. Excellent EAs for micafungin (100%), fluconazole (90.0%), and voriconazole (95.0%) were found for *C. glabrata*. Although fluconazole and caspofungin in *C. glabrata* are not recommended to be read with YS08, we compared the MIC distributions between the two methods (Fig. 1). For fluconazole, the MICs by BMD were more widely distributed than those by YS08, and for caspofungin, YS08 generated at least a 2-fold higher MIC compared with BMD.

In *C. krusei*, about 26.7% (4/15) of isolates in BMD and 80% (12/15) of isolates in YS08 showed intermediate susceptibility to caspofungin, but all showed micafungin susceptibility in BMD and YS08. *C. parapsilosis* showed 5.3% (1/19) of very major errors and 10.5% (2/19) of minor errors for fluconazole and 5.3% of minor errors for caspofungin. In *C. tropicalis*, 33.3% (7/21) of minor errors and no very major errors or major errors were observed for fluconazole. The EAs between the BMD and YS08 were 76.2%, 100%, 100%, 100%, and 100% for fluconazole, voriconazole, micafungin, caspofungin, and amphotericin B, respectively, in *C. tropicalis*.

The EAs of flucytosine were 100% for all strains except *C. albicans* (87.5%, 21/24). Because the minimum MIC value that can be reported by the YS08 for flucytosine is 1 $\mu$g/mL, MICs less than 1 $\mu$g/mL could not be reported in YS08 (Table S1 in the supplemental material).

We compared the fluconazole MICs and voriconazole MICs in *C. glabrata* to predict the fluconazole susceptibility using different criteria (Table 2). The CAs between BMD fluconazole MICs and YS08 voriconazole MICs were 90% (18/20) by the method-specific CBP of voriconazole, 80% (16/20) by the current ECV (0.25 $\mu$g/mL) of voriconazole, and 85% (17/20) by the previous ECV (0.5 $\mu$g/mL) of voriconazole. Only 10% of very major errors were detected by the method-specific CBP. About 5% of very major errors and 15% of major errors were detected by the current ECV criteria, whereas 10% of very major errors and 5% of major errors were found by the previous ECV criteria. However, the CAs between fluconazole ECV (8 $\mu$g/mL) and YS08 voriconazole ECV were 65% (13/20) by the current ECV criteria and 60% (12/20) by the previous ECV criteria.

In Table 3, the proportion of WT versus non-WT and CAs were analyzed according to the CLSI ECV criteria. Agreement between voriconazole and micafungin could not be determined in *C. albicans* because the minimum MIC reported by YS08 was higher than the ECV concentration (ECV: 0.03 for voriconazole and micafungin). Micafungin for *C. glabrata* could also not be interpreted for the same reason. The CA of *C. glabrata* for voriconazole was 80% (16/20), of which two non-WT isolates were classified as WT in YS08. In *C. tropicalis*, the CA for fluconazole was 38.1% and that for voriconazole was 66.7%. Thirteen (61.9%) non-WT isolates for fluconazole and seven (33.3%) for voriconazole were classified in YS08 as WT, respectively.

Based on the Centers for Disease Control and Prevention (CDC) criteria, *C. auris* showed 37.8% (17/45) fluconazole resistance by BMD and 22.2% (10/45) by YS08 (Table 4). Only 24.4% (11/45) of isolates showed amphotericin B resistance in BMD. No

**TABLE 1** *In vitro* antifungal susceptibility profiles analyzed by CLSI clinical breakpoints except for *C. auris*[a]

| Candida species | Antifungal agents | Method | Breakpoints (µg/mL) | MIC (mg/L) Median (range) | EA n (%) | S n (%) | I or SDD n (%) | R n (%) | CA n (%) | VME n (%) | ME n (%) | MiE n (%) |
|---|---|---|---|---|---|---|---|---|---|---|---|---|
| *C. albicans* (n = 24) | Fluconazole | BMD | S: ≤ 2, SDD: 4, R: ≥ 8 | 0.5 (0.12–32) | 20 (83.3) | 18 (75) | 2 (8.3) | 4 (16.7) | 19 (79.2) | 3 (12.5) | | 2 (8.3) |
| | | YS08 | | 0.5 (0.5–8) | | 23 (95.8) | | 1 (4.2) | | | | |
| | Voriconazole | BMD | S:≤ 0.12, I: 0.25–0.5, R: ≥ 1 | 0.03 (0.008–16) | 22 (91.7) | 22 (91.7) | 1 (4.2) | 1 (4.2) | 22 (91.7) | 1 (4.2) | 1 (4.2) | |
| | | YS08 | | 0.12 (0.06–4) | | 22 (91.7) | 1 (4.2) | 1 (4.2) | | | | |
| | Micafungin | BMD | S: ≤ 0.25, I: 0.5, R: ≥ 1 | 0.015 (0.008–8) | 23 (95.8) | 23 (95.8) | | | 23 (95.8) | 1 (4.2) | | |
| | | YS08 | | 0.06 (0.06) | | 24 (100) | | | | | | |
| | Caspofungin | BMD | S: ≤ 0.25, I: 0.5, R: ≥ 1 | 0.06 (0.008–8) | 23 (95.8) | 23 (95.8) | | 1 (4.2) | 23 (95.8) | 1 (4.2) | | |
| | | YS08 | | 0.12 (0.12) | | 24 (100) | | | | | | |
| | Amphotericin B | BMD | NA | 0.5 (0.25–8) | 23 (95.8) | | | | | | | |
| | | YS08 | NA | 0.5 (0.25–0.5) | | | | | | | | |
| | Flucytosine | BMD | NA | 0.06 (0.06–64) | 21 (87.5) | | | | | | | |
| | | YS08 | NA | 1 (1–64) | | | | | | | | |
| *C. glabrata* (n = 20) | Fluconazole | BMD | SDD:≤ 32, R: ≥ 64 | 16 (2–64) | 18 (90) | | 15 (75) | 5 (25) | | | | |
| | | YS08 | | 16 (8–32) | | | | | | | | |
| | Voriconazole | BMD | NA | 0.25 (0.03–8) | 19 (95) | | | | | | | |
| | | YS08[b] | S: ≤ 1, I: 2, R: ≥ 4 | 0.25 (0.12–8) | | 20 (100) | | | | | | |
| | Micafungin | BMD | S: ≤ 0.06, I: 0.12, R: ≥ 0.25 | 0.015 (0.008–0.06) | 20 (100) | 16 (80) | 1 (5.0) | 3 (15) | 20 (100) | | | |
| | | YS08 | | 0.06 (0.06) | | 20 (100) | | | | | | |
| | Caspofungin | BMD | S: ≤ 0.12, I: 0.25, R: ≥ 0.5 | 0.03 (0.03–0.25) | 9 (45) | 3 (15) | 15 (75) | 2 (10) | 3 (15) | 0 (0) | 2 (10) | 15 (75) |
| | | YS08 | | 0.25 (0.12–5) | | 3 (15) | | | | | | |
| | Amphotericin B | BMD | NA | 1 (0.12–2) | 20 (100) | | | | | | | |
| | | YS08 | NA | 0.5 (0.5–1) | | | | | | | | |
| | Flucytosine | BMD | NA | 0.06 (0.06) | 20 (100) | | | | | | | |
| | | YS08 | NA | 1 (1) | | | | | | | | |
| *C. guilliermondii* (n = 9) | Fluconazole | BMD | NA | 4 (1–16) | 8 (88.9) | | | | | | | |
| | | YS08 | NA | 2 (2–4) | | | | | | | | |
| | Voriconazole | BMD | NA | 0.06 (0.015–0.5) | 9 (100) | | | | | | | |
| | | YS08 | NA | 0.12 (0.12–0.25) | | | | | | | | |
| | Micafungin | BMD | S: ≤ 2, I: 4, R: ≥ 8 | 0.5 (0.06–2) | 7 (77.8) | 9 (100) | | | 9 (100) | | | |
| | | YS08 | | 0.5 (0.12–2) | | 9 (100) | | | | | | |
| | Caspofungin | BMD | S: ≤ 2, I: 4, R: ≥ 8 | 0.5 (0.06–2) | 8 (88.9) | 9 (100) | | | 9 (100) | | | |
| | | YS08 | | 0.5 (0.25–1) | | 9 (100) | | | | | | |
| | Amphotericin B | BMD | NA | 0.5 (0.12–0.5) | 9 (100) | | | | | | | |
| | | YS08 | NA | 0.25 (0.25–0.5) | | | | | | | | |
| | Flucytosine | BMD | NA | 0.06 (0.06–0.12) | 9 (100) | | | | | | | |
| | | YS08 | NA | 1 (1) | | | | | | | | |
| *C. krusei* (n = 15) | Fluconazole | BMD | NA | 64 (32–128) | 6 (40) | | | | | | | |
| | | YS08 | NA | 8 (8) | | | | | | | | |
| | Voriconazole | BMD | S: ≤ 0.5, I: 1, R: ≥ 2 | 0.25 (0.25–0.5) | 15 (100) | 15 (100) | | | 15 (100) | | | |
| | | YS08 | | 0.12 (0.12–0.25) | | 15 (100) | | | | | | |
| | Micafungin | BMD | S: ≤ 0.25, I: 0.5, R: ≥ 1 | 0.12 (0.06–0.25) | 15 (100) | 15 (100) | | | 15 (100) | | | |
| | | YS08 | | 0.12 (0.06–0.12) | | 15 (100) | | | | | | |
| | Caspofungin | BMD | S: ≤ 0.25, I: 0.5, R: ≥ 1 | 0.25 (0.12–0.5) | 15 (100) | 11 (73.3) | 4 (26.7) | | 7 (46.7) | | | 8 (53.3) |
| | | YS08 | | 0.5 (0.25–0.5) | | 3 (20) | 12 (80) | | | | | |
| | Amphotericin B | BMD | NA | 1 (0.5–2) | 14 (93.3) | | | | | | | |
| | | YS08 | NA | 0.5 (0.25–4) | | | | | | | | |
| | Flucytosine | BMD | NA | 16 (8–32) | 15 (100) | | | | | | | |
| | | YS08 | NA | 16 (8–32) | | | | | | | | |

**TABLE 1** (Continued)

| Candida species | Antifungal agents | Method | Breakpoints (µg/mL) | MIC (mg/L) Median (range) | EA n (%) | S n (%) | I or SDD n (%) | R n (%) | CA n (%) | VME n (%) | ME n (%) | MiE n (%) |
|---|---|---|---|---|---|---|---|---|---|---|---|---|
| C. lusitaniae (n = 6)[c] | Fluconazole | BMD | S: ≤ 2, SDD:4, R: ≥ 8 | 0.5 (0.25–1) | 6 (100) | 6 (100) | | | 6 (100) | | | |
| | | YS08 | | 0.5 (0.5) | | 6 (100) | | | | | | |
| | Voriconazole | BMD | S: ≤ 0.12, I: 0.25–0.5, R: ≥ 1 | 0.008 (0.008–0.015) | 6 (100) | 6 (100) | | | 6 (100) | | | |
| | | YS08 | | 0.12 (0.12) | | 6 (100) | | | | | | |
| | Micafungin | BMD | S: ≤ 0.25, I: 0.5, R: ≥ 1 | 0.06 (0.03–0.06) | 6 (100) | 6 (100) | | | 6 (100) | | | |
| | | YS08 | | 0.12 (0.12) | | 6 (100) | | | | | | |
| | Caspofungin | BMD | S: ≤ 0.25, I: 0.5, R: ≥ 1 | 0.185 (0.06–0.25) | 6 (100) | 6 (100) | | | 6 (100) | | | |
| | | YS08 | | 0.25 (0.25) | | 6 (100) | | | | | | |
| | Amphotericin B | BMD | NA | 0.25 (0.25–0.5) | 6 (100) | | | | | | | |
| | | YS08 | | 0.5 (0.5) | | | | | | | | |
| | Flucytosine | BMD | NA | 0.06 (0.06) | 6 (100) | | | | | | | |
| | | YS08 | | 1 (1) | | | | | | | | |
| C. orthopsilosis (n = 5)[c] | Fluconazole | BMD | S: ≤ 2, SDD: 4, R: ≥ 8 | 1 (0.5–4) | 4 (80) | 4 (80) | 1 (20) | | 4 (80) | | | 1 (20) |
| | | YS08 | | 0.5 (0.5) | | 5 (100) | | | | | | |
| | Voriconazole | BMD | S: ≤ 0.12, I: 0.25–0.5, R: ≥ 1 | 0.03 (0.015–0.25) | 5 (100) | 4 (80) | 1 (20) | | 4 (80) | | | 1 (20) |
| | | YS08 | | 0.12 (0.12) | | 5 (100) | | | | | | |
| | Micafungin | BMD | S: ≤ 0.25, I: 0.5, R: ≥ 1 | 0.5 (0.5) | 5 (100) | | 5 (100) | | 1 (20) | | | 4 (80) |
| | | YS08 | | 0.12 (0.12–0.5) | | 4 (80) | 1 (20) | | | | | |
| | Caspofungin | BMD | S: ≤ 0.25, I: 0.5, R: ≥ 1 | 0.5 (0.5) | 5 (100) | | 5 (100) | | 0 | | | 5 (100) |
| | | YS08 | | 0.12 (0.12–0.25) | | 5 (100) | | | | | | |
| | Amphotericin B | BMD | NA | 0.5 (0.5–1) | 5 (100) | | | | | | | |
| | | YS08 | | 0.25 (0.25–0.5) | | | | | | | | |
| | Flucytosine | BMD | NA | 0.12 (0.06–0.5) | 5 (100) | | | | | | | |
| | | YS08 | | 1 (1) | | | | | | | | |
| C. parapsilosis (n = 19) | Fluconazole | BMD | S: ≤ 2, SDD: 4, R: ≥ 8 | 2 (0.25–8) | 17 (89.5) | 17 (89.5) | 1 (5.3) | 1 (5.3) | 16 (84.2) | 1 (5.3) | | 2 (10.5) |
| | | YS08 | | 0.5 (0.5–4) | | 18 (94.7) | 1 (5.3) | | | | | |
| | Voriconazole | BMD | S: ≤ 0.12, I: 0.25–0.5, R: ≥ 1 | 0.03 (0.008–0.06) | 19 (100) | 19 (100) | | | 19 (100) | | | |
| | | YS08 | | 0.12 (0.12) | | 19 (100) | | | | | | |
| | Micafungin | BMD | S: ≤ 2, I: 4, R: ≥ 8 | 2 (1–2) | 19 (100) | 19 (100) | | | 19 (100) | | | |
| | | YS08 | | 0.5 (0.5–1) | | 19 (100) | | | | | | |
| | Caspofungin | BMD | S: ≤ 2, I: 4, R: ≥ 8 | 1 (0.5–4) | 19 (100) | 18 (94.7) | 1 (5.3) | | 18 (94.7) | | | 1 (5.3) |
| | | YS08 | | 1 (0.5–1) | | 19 (100) | | | | | | |
| | Amphotericin B | BMD | NA | 0.5 (0.12–1) | 19 (100) | | | | | | | |
| | | YS08 | | 0.5 (0.5–1) | | | | | | | | |
| | Flucytosine | BMD | NA | 0.06 (0.06–0.25) | 19 (100) | | | | | | | |
| | | YS08 | | 1 (1) | | | | | | | | |
| C. pelliculosa (n = 8)[c] | Fluconazole | BMD | S: ≤ 2, SDD: 4, R: ≥ 8 | 4 (2–8) | 8 (100) | 2 (25) | 5 (62.5) | 1 (12.5) | 2 (25) | 1 (12.5) | | 5 (62.5) |
| | | YS08 | | 2 (2) | | 8 (100) | | | | | | |
| | Voriconazole | BMD | S: ≤ 0.12, I: 0.25–0.5, R: ≥ 1 | 0.12 (0.06–0.25) | 8 (100) | 7 (87.5) | 1 (12.5) | | 8 (100) | | | |
| | | YS08 | | 0.12 (0.12–0.25) | | 7 (87.5) | 1 (12.5) | | | | | |
| | Micafungin | BMD | S: ≤ 0.25, I: 0.5, R: ≥ 1 | 0.03 (0.015–0.03) | 8 (100) | 8 (100) | | | 8 (100) | | | |
| | | YS08 | | 0.06 (0.06) | | 8 (100) | | | | | | |
| | Caspofungin | BMD | S: ≤ 0.25, I: 0.5, R: ≥ 1 | 0.03 (0.015–0.06) | 7 (87.5) | 8 (100) | | | 8 (100) | | | |
| | | YS08 | | 0.185 (0.12–0.25) | | 8 (100) | | | | | | |
| | Amphotericin B | BMD | NA | 0.25 (0.12–1) | 8 (100) | | | | | | | |
| | | YS08 | | 0.5 (0.25–0.5) | | | | | | | | |
| | Flucytosine | BMD | NA | 0.06 (0.06) | 8 (100) | | | | | | | |
| | | YS08 | | 1 (1) | | | | | | | | |

**TABLE 1** (Continued)

| Candida species | Antifungal agents | Method | Breakpoints (μg/mL) | MIC (mg/L) Median (range) | EA n (%) | S n (%) | I or SDD n (%) | R n (%) | CA n (%) | VME n (%) | ME n (%) | MiE n (%) |
|---|---|---|---|---|---|---|---|---|---|---|---|---|
| C. tropicalis (n = 21) | Fluconazole | BMD | S: ≤ 2, SDD: 4, R: ≥ 8 | 2 (0.5–4) | 16 (76.2) | 14 (66.7) | 7 (33.3) | | 14 (66.7) | | | 7 (33.3) |
| | | YS08 | | 0.5 (0.5–1) | | 21 (100) | | | | | | |
| | Voriconazole | BMD | S: ≤ 0.12, I: 0.25–0.5, R: ≥ 1 | 0.06 (0.03–0.12) | 21 (100) | 21 (100) | | | 21 (100) | | | |
| | | YS08 | | 0.12 (0.12) | | 21 (100) | | | | | | |
| | Micafungin | BMD | S: ≤ 0.25, I: 0.5, R: ≥ 1 | 0.03 (0.015–0.03) | 21 (100) | 21 (100) | | | 21 (100) | | | |
| | | YS08 | | 0.06 (0.06) | | 21 (100) | | | | | | |
| | Caspofungin | BMD | S: ≤ 0.25, I: 0.5, R: ≥ 1 | 0.06 (0.03–0.25) | 21 (100) | 21 (100) | | | 21 (100) | | | |
| | | YS08 | | 0.12 (0.12–0.25) | | 21 (100) | | | | | | |
| | Amphotericin B | BMD | NA | 1 (0.5–2) | 21 (100) | | | | | | | |
| | | YS08 | | 0.5 (0.25–0.5) | | | | | | | | |
| | Flucytosine | BMD | NA | 0.06 (0.06–0.12) | 21 (100) | | | | | | | |
| | | YS08 | | 1 (1) | | | | | | | | |
| C. fabianii (n = 12)[c] | Fluconazole | BMD | S: ≤ 2, SDD: 4, R: ≥ 8 | 2 (1–4) | 12 (100) | 10 (83.3) | 2 (16.7) | | 10 (83.3) | | | 2 (16.7) |
| | | YS08 | | 0.5 (0.5–2) | | 12 (100) | | | | | | |
| | Voriconazole | BMD | S: ≤ 0.12, I: 0.25–0.5, R: ≥ 1 | 0.06 (0.015–0.25) | 11 (91.7) | 12 (100) | | | 12 (100) | | | |
| | | YS08 | | 0.12 (0.12) | | 12 (100) | | | | | | |
| | Micafungin | BMD | S: ≤ 0.25, I: 0.5, R: ≥ 1 | 0.06 (0.03–0.12) | 12 (100) | 12 (100) | | | 12 (100) | | | |
| | | YS08 | | 0.12 (0.06–0.12) | | 12 (100) | | | | | | |
| | Caspofungin | BMD | S: ≤ 0.25, I: 0.5, R: ≥ 1 | 0.12 (0.03–0.25) | 10 (83.3) | 12 (100) | | | 12 (100) | | | |
| | | YS08 | | 0.25 (0.12–0.25) | | 12 (100) | | | | | | |
| | Amphotericin B | BMD | NA | 1 (0.25–1) | 12 (100) | | | | | | | |
| | | YS08 | | 0.25 (0.25–0.5) | | | | | | | | |
| | Flucytosine | BMD | NA | 0.12 (0.06–0.25) | 12 (100) | | | | | | | |
| | | YS08 | | 1 (1) | | | | | | | | |

[a]EA, essential agreement; S, susceptible; I, intermediate; SDD, susceptible dose-dependent; R, resistant; CA, categorical agreement; VME, very major errors; ME, major errors; MiE, minor errors; BMD, CLSI broth microdilution method; YS08, Vitek 2 AST YS08; NA, not available.

[b]Method-specific CBP of YS08 voriconazole was used.

[c]Because CBPs for C. lusitaniae, C. orthopsilosis, C. pelliculosa and C. fabianii were not available, the value from C. albicans was used to identify isolates/species with elevated MICs.

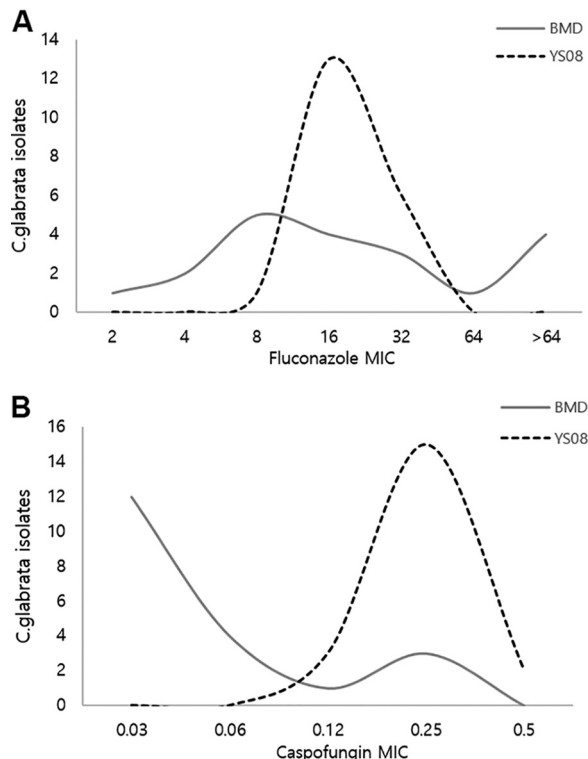

**FIG 1** Comparison of fluconazole (A) and caspofungin (B) MIC distributions in BMD and YS08.

isolates showed resistance to micafungin or caspofungin. YS08 MIC50 values were within one dilution of BMD, except for flucytosine. YS08 MIC90 values were within two dilutions of BMD, except for micafungin. The EAs between the CLSI and YS08 were 93.3%, 100%, 82.2%, 97.8%, 97.8%, and 97.8% for fluconazole, voriconazole, micafungin, caspofungin, amphotericin B, and flucytosine, respectively.

## DISCUSSION

In this study, we report six antifungal susceptibility profiles for 184 clinical isolates recovered in Korea. The CBPs were used to define susceptible or resistant isolates. ECVs

**TABLE 2** Comparison of fluconazole and voriconazole MICs obtained by CLSI broth microdilution method and Vitek 2 AST–YS08 of *C. glabrata*[a]

| | | Results of YS08 for voriconazole | | | | | |
|---|---|---|---|---|---|---|---|
| | | Method-specific CBP from YS08 (S: ≤1, I: 2, R: 4 $\mu$g/mL) | | Current ECV (0.25 $\mu$g/mL) | | Previous ECV (0.5 $\mu$g/mL) | |
| **Results of BMD for fluconazole** | | **S + I** | **R** | **WT** | **Non–WT** | **WT** | **Non–WT** |
| CBP (SDD: ≤ 32, R: ≥ 64 $\mu$g/mL) | SDD (n = 15) | 15 | 0 | 12 | 3 | 14 | 1 |
| | R (n = 5) | 2 | 3 | 1 | 4 | 2 | 3 |
| | CA, n (%) | 18 (90) | | 16 (80) | | 17 (85) | |
| | VME, n (%) | 2 (10) | | 1 (5) | | 2 (10) | |
| | ME, n (%) | 0 | | 3 (15) | | 1 (5) | |
| ECV (WT: ≤ 8, non-WT : >8 $\mu$g/mL) | WT (n = 8) | 8 | 0 | 7 | 1 | 8 | 0 |
| | Non–WT (n = 12) | 9 | 3 | 6 | 6 | 8 | 4 |
| | CA, n (%) | 11 (55) | | 13 (65) | | 12 (60) | |
| | VME, n (%) | 9 (45) | | 6 (30) | | 8 (40) | |
| | ME, n (%) | 0 | | 1 (5) | | 0 | |

[a]S, susceptible; I, intermediate; SDD, susceptible dose-dependent; R, resistant; CA, categorical agreement; VME, very major errors; ME, major errors; WT, wild type; non-WT, non-wild type.

**TABLE 3** *In vitro* antifungal susceptibility profiles analyzed by CLSI epidemiological cutoff values except for *C. auris*[a]

| *Candida* species | Antifungal agents | Method | Breakpoints ($\mu$g/mL) | Wild type N, (%) | Non–wild type N, (%) | CA N, (%) |
|---|---|---|---|---|---|---|
| *C. albicans* (*n* = 24) | Fluconazole | BMD | ECV: 0.5 | 19 (79.2) | 5 (20.8) | 21 (87.5) |
| | | YS08 | | 18 (75) | 6 (25) | |
| | Voriconazole | BMD | ECV: 0.03 | 20 (83.3) | 4 (16.7) | |
| | | YS08 | | NA | NA | |
| | Micafungin | BMD | ECV: 0.03 | 23 (95.8) | 1 (4.2) | |
| | | YS08 | | NA | NA | |
| | Caspofungin | BMD | NA | | | |
| | | YS08 | | | | |
| | Amphotericin B | BMD | ECV: 2 | 23 (95.8) | 1 (4.2) | 23 (95.8) |
| | | YS08 | | 24 (100) | | |
| | Flucytosine | BMD | ECV: 1 | 16 (66.7) | 8 (33.3) | 22 (91.7) |
| | | YS08 | | 16 (66.7) | 8 (33.3) | |
| *C. glabrata* (*n* = 20) | Fluconazole | BMD | ECV: 8 | 8 (40) | 12 (60) | |
| | | YS08 | | | | |
| | Voriconazole | BMD | ECV: 0.25 | 11 (55) | 9 (45) | 16 (80) |
| | | YS08 | | 13 (65) | 7 (35) | |
| | Micafungin | BMD | ECV: 0.03 | 19 (95) | 1 (5) | |
| | | YS08 | | NA | NA | |
| | Caspofungin | BMD | | | | |
| | | YS08 | | | | |
| | Amphotericin B | BMD | ECV: 2 | 20 (100) | | 20 (100) |
| | | YS08 | | 20 (100) | | |
| | Flucytosine | BMD | ECV: 1 | 20 (100) | | 20 (100) |
| | | YS08 | | 20 (100) | | |
| *C. guilliermondii* (*n* = 9) | Fluconazole | BMD | ECV: 8 | 8 (88.9) | 1 (11.1) | 8 (88.9) |
| | | YS08 | | 9 (100) | | |
| | Voriconazole | BMD | NA | | | |
| | | YS08 | | | | |
| | Micafungin | BMD | ECV: 2 | 9 (100) | | 9 (100) |
| | | YS08 | | 9 (100) | | |
| | Caspofungin | BMD | ECV: 2 | 9 (100) | | 9 (100) |
| | | YS08 | | 9 (100) | | |
| | Amphotericin B | BMD | ECV: 2 | 9 (100) | | 9 (100) |
| | | YS08 | | 9 (100) | | |
| | Flucytosine | BMD | ECV: 1 | 9 (100) | | 9 (100) |
| | | YS08 | | 9 (100) | | |
| *C. krusei* (*n* = 15) | Fluconazole | BMD | NA | | | |
| | | YS08 | | | | |
| | Voriconazole | BMD | ECV: 0.5 | 15 (100) | | 15 (100) |
| | | YS08 | | 15 (100) | | |
| | Micafungin | BMD | ECV: 0.25 | 15 (100) | | 15 (100) |
| | | YS08 | | 15 (100) | | |
| | Caspofungin | BMD | NA | | | |
| | | YS08 | | | | |
| | Amphotericin B | BMD | ECV: 2 | 15 (100) | | 14 (93.3) |
| | | YS08 | | 14 (93.3) | 1 (6.7) | |
| | Flucytosine | BMD | ECV: 1 | | 15 (100) | 15 (100) |
| | | YS08 | | | 15 (100) | |
| *C. lusitaniae* (*n* = 6) | Fluconazole | BMD | ECV: 1 | 6 (100) | | 6 (100) |
| | | YS08 | | 6 (100 | | |
| | Voriconazole | BMD | NA | | | |
| | | YS08 | | | | |
| | Micafungin | BMD | ECV: 0.5 | 6 (100 | | 6 (100) |
| | | YS08 | | 6 (100 | | |
| | Caspofungin | BMD | ECV: 1 | 6 (100 | | 6 (100) |
| | | YS08 | | 6 (100 | | |
| | Amphotericin B | BMD | ECV: 2 | 6 (100 | | 6 (100) |
| | | YS08 | | 6 (100 | | |
| | Flucytosine | BMD | ECV: 1 | 6 (100 | | 6 (100) |
| | | YS08 | | 6 (100 | | |

**TABLE 3** (Continued)

| *Candida* species | Antifungal agents | Method | Breakpoints (µg/mL) | Wild type N, (%) | Non–wild type N, (%) | CA N, (%) |
|---|---|---|---|---|---|---|
| *C. orthopsilosis* (n = 5) | Fluconazole | BMD | ECV: 2 | 4 (80) | 1 (20) | 4 (80) |
| | | YS08 | | 5 (100) | | |
| | Voriconazole | BMD | ECV: 0.125 | 4 (80) | 1 (20) | 4 (80) |
| | | YS08 | | 5 (100) | | |
| | Micafungin | BMD | ECV: 1 | 5 (100) | | 5 (100) |
| | | YS08 | | 5 (100) | | |
| | Caspofungin | BMD | ECV: 1 | 5 (100) | | 5 (100) |
| | | YS08 | | 5 (100) | | |
| | Amphotericin B | BMD | ECV: 2 | 5 (100) | | 5 (100) |
| | | YS08 | | 5 (100) | | |
| | Flucytosine | BMD | ECV: 1 | 5 (100) | | 5 (100) |
| | | YS08 | | 5 (100) | | |
| *C. parapsilosis* (n = 19) | Fluconazole | BMD | ECV: 2 | 17 (89.5) | 2 (10.5) | 16 (84.2) |
| | | YS08 | | 18 (94.7) | 1 (5.3) | |
| | Voriconazole | BMD | NA | | | |
| | | YS08 | | | | |
| | Micafungin | BMD | ECV: 2 | 19 (100) | | 19 (100) |
| | | YS08 | | 19 (100) | | |
| | Caspofungin | BMD | ECV: 1 | 18 (94.7) | 1 (5.3) | 18 (94.7) |
| | | YS08 | | 19 (100) | | |
| | Amphotericin B | BMD | ECV: 1 | 19 (100) | | 19 (100) |
| | | YS08 | | 19 (100) | | |
| | Flucytosine | BMD | ECV: 1 | 19 (100) | | 19 (100) |
| | | YS08 | | 19 (100) | | |
| *C. pelliculosa* (n = 8)[b] | Fluconazole | BMD | ECV: 0.5 | | 8 (100) | 8 (100) |
| | | YS08 | | | 8 (100) | |
| | Voriconazole | BMD | ECV: 0.03 | | 8 (100) | |
| | | YS08 | | NA | | |
| | Micafungin | BMD | ECV: 0.03 | 8 (100) | | |
| | | YS08 | | NA | | |
| | Caspofungin | BMD | NA | | | |
| | | YS08 | | | | |
| | Amphotericin B | BMD | ECV: 2 | 8 (100) | | 8 (100) |
| | | YS08 | | 8 (100) | | |
| | Flucytosine | BMD | ECV: 1 | 8 (100) | | 8 (100) |
| | | YS08 | | 8 (100) | | |
| *C. tropicalis* (n = 21) | Fluconazole | BMD | ECV: 1 | 8 (38.1) | 13 (61.9) | 8 (38.1) |
| | | YS08 | | 21 (100) | | |
| | Voriconazole | BMD | ECV: 0.12 | 14 (66.7) | 7 (33.3) | 14 (66.7) |
| | | YS08 | | 21 (100) | | |
| | Micafungin | BMD | ECV: 0.06 | 21 (100) | | 21 (100) |
| | | YS08 | | 21 (100) | | |
| | Caspofungin | BMD | NA | | | |
| | | YS08 | | | | |
| | Amphotericin B | BMD | ECV: 2 | 21 (100) | | 21 (100) |
| | | YS08 | | 21 (100) | | |
| | Flucytosine | BMD | ECV: 1 | 21 (100) | | 21 (100) |
| | | YS08 | | 21 (100) | | |
| *C. fabianii* (n = 12)[b] | Fluconazole | BMD | ECV: 0.5 | 0 | 13 (100) | 2 (16.7) |
| | | YS08 | | 10 (83.3) | 2 (16.7) | |
| | Voriconazole | BMD | ECV: 0.03 | 6 (50.0) | 6 (50.0) | |
| | | YS08 | | NA | | |
| | Micafungin | BMD | ECV: 0.03 | 5 (41.7) | 7 (58.3) | |
| | | YS08 | | NA | | |
| | Caspofungin | BMD | NA | | | |
| | | YS08 | | | | |
| | Amphotericin B | BMD | ECV: 2 | 12 (100) | | 12(100) |
| | | YS08 | | 12(100) | | |
| | Flucytosine | BMD | ECV: 1 | 12(100) | | 12(100) |
| | | YS08 | | 12(100) | | |

[a]CA, categorical agreement; BMD, CLSI broth microdilution method; YS08, Vitek 2 AST YS08; NA, not available.
[b]Because ECVs for *C. pelliculosa* and *C. fabianii* were not available, the value from *C. albicans* was used to identify isolates/species with elevated MICs.

**TABLE 4** MIC distributions of *C. auris* (*n* = 45)[a]

| Antifungal agents | Test method | MIC (mg/L) | | | | | | | | | | | | | Median | EA (*n*, %) | MIC50 | MIC90 |
|---|---|---|---|---|---|---|---|---|---|---|---|---|---|---|---|---|---|---|
| | | ≤0.015 | 0.03 | 0.06 | 0.12 | 0.25 | 0.5 | 1 | 2 | 4 | 8 | 16 | 32 | >64 | | | | |
| Fluconazole | BMD | | | | | | | | 7 | 4 | 7 | 10 | 8 | 9 | 16 | 42 (93.3%) | 16 | 64 |
| | YS08 | | | | | | 0 | | 3 | 2 | 26 | 4 | 10 | | 8 | | 8 | 32 |
| Voriconazole | BMD | 4 | 10 | 12 | 3 | 7 | 7 | 1 | 1 | | | | | | 0.06 | 45 (100%) | 0.06 | 0.5 |
| | YS08 | | | | 29 | 8 | 7 | 1 | | | | | | | 0.12 | | 0.12 | 0.5 |
| Micafungin | BMD | | 8 | 13 | 13 | 3 | 6 | 2 | | | | | | | 0.12 | 37 (82.2%) | 0.12 | 0.5 |
| | YS08 | | | 43 | 2 | | | | | | | | | | 0.06 | | 0.06 | 0.06 |
| Caspofungin | BMD | | 3 | 11 | 21 | 8 | 2 | | | | | | | | 0.12 | 44 (97.8%) | 0.12 | 0.25 |
| | YS08 | | | | 20 | 25 | | | | | | | | | 0.25 | | 0.25 | 0.25 |
| Amphotericin B | BMD | | | | | 7 | 19 | 8 | 11 | | | | | | 0.5 | 44 (97.8%) | 0.5 | 2.0 |
| | YS08 | | | | | 17 | 28 | | | | | | | | 0.5 | | 0.5 | 0.5 |
| 5–Flucytosine | BMD | | | 11 | 19 | 13 | 2 | | | | | | | | 0.12 | 44 (97.8%) | 0.12 | 0.25 |
| | YS08 | | | | | | | 44 | | | 1 | | | | 1 | | 1 | 1 |

[a]BMD, CLSI broth microdilution method; YS08, Vitek 2 AST YS08. Isolate numbers tested are given in parentheses for each species. Species above the vertical line are resistance strains according to the CDC recommendation (fluconazole ≥32 $\mu$g/mL; amphotericin B ≥2 $\mu$g/mL; caspofungin ≥2 $\mu$g/mL; micafungin ≥4 $\mu$g/mL). Gray color: Low off–scale MIC value of Vitek 2 AST YS08.

are useful in distinguishing between WT without resistance mechanisms and non-WT with resistance mechanisms (15). Both criteria can be used to obtain the CA between the two methods (17). Currently, two commercial BMD-based systems, the Vitek 2 system (bioMérieux, France) and the Sensititre YeastOne system (Thermo Scientific, Cleveland, OH, USA), are widely used in clinical microbiological laboratories for antifungal susceptibility testing (17). The previous version of the YS08, YS07, had an FDA-accredited fluconazole formulation that has been validated against *C. glabrata*. However, the fluconazole formulation was modified and no validation for *C. glabrata* has been made; therefore, the MIC for the fluconazole of *C. glabrata* is not recorded in YS08 (18).

The azoles are the most commonly used agents for invasive fungal infections, in particular, fluconazole for the treatment of candidemia (2). Regardless of the *Candida* species tested, the overall EA and CA of fluconazole between BMD and YS08 ranged from 40%–100% and from 66.7%–100%, respectively. Notably, YS08 classified five fluconazole-resistant isolates as susceptible, representing a very major error. *C. albicans*, *C. parapsilosis*, and *C. tropicalis* showed less than 90% EA and CA for fluconazole MIC. *C. albicans* showed 79.2% CA and 83.3% EA, and 12.5% of very major errors in fluconazole were detected by YS08. *C. parapsilosis* showed 84.2% CA and 89.5% EA with 5.3% of very major errors for fluconazole. In *C. tropicalis*, 66.7% CA, 66.7% EA, and 33.3% of minor errors were detected, and the median MIC (range) for fluconazole was 2 (0.5–4) using the CLSI method and 0.5 (0.5–1) by YS08. These findings led to the conclusion that the fluconazole MIC of YS08 was lower than that of the BMD method. Until recently, only two studies had assessed the clinical performance of YS08 cards compared with Sensititre YeastOne and/or BMD using clinical isolates. Comparing BMD and YS08, Lim et al. found excellent (>90%) EA and CA for fluconazole in *C. albicans* and *C. parapsilosis*, but 3.8% of very major errors and 13% of minor errors were observed in *C. albicans* and *C tropicalis* (17). However, in another study comparing YS08 and Sensititre YeastOne, Wong et al. revealed poor agreement for fluconazole in *C. albicans*, with 70% EA and 85% CA (19). The better and more consistent EA and CA results compared with our study might be due to the *Candida* species tested.

In our study, the EA of *C. glabrata* against fluconazole and voriconazole was excellent (>90%), but a different MIC distribution for fluconazole was observed between BMD and YS08. The CLSI guideline does not provide the voriconazole MIC value because of insufficient data on the relationship between *C. glabrata* and voriconazole resistance in clinical outcomes (14). However, the cross-resistance between fluconazole and voriconazole has been reported (20). In our study, the CAs with BMD fluconazole and YS08 voriconazole were 90%, 85%, and 80% using the method-specific CBP, ECV of 0.5 $\mu$g/mL, and ECV of 0.25 $\mu$g/mL, respectively. Lim et al. reported 97.8% and 82.5% CA between BMD fluconazole MIC and ECV of 0.5 $\mu$g/mL and 0.25 $\mu$g/mL, respectively (17). Therefore, YS08

voriconazole testing using the CLSI ECV might provide reliable discrimination of fluconazole-resistant *C. glabrata*.

The CLSI recommends caution in interpretation because antifungal susceptibility tests for caspofungin have high interlaboratory variability (14). Caspofungin can show "eagle effects," which may cause variability in MICs, as described previously (21). When susceptibility tests are performed under standardized conditions, a surprising echinocandin-specific paradoxical effect can be observed. This effect refers to a phenomenon in which certain strains grow in higher concentrations of echinocandin while being completely susceptible at lower concentrations. This has been observed in several species of *Candida* and *Aspergillus* (22).

*C. glabrata* and *C. krusei* showed excellent CA for micafungin (100% and 100%, respectively) but poor CA (15% and 46.7%, respectively) for caspofungin. In *C. krusei*, 26.7% (4/15) of isolates in BMD showed intermediate susceptibility to caspofungin but were susceptible to micafungin. Although the reference method used was the CLSI BMD, caspofungin susceptibility testing *in vitro* might have contributed to reports of false resistance (23). Therefore, our results of a low EA for caspofungin of *C. glabrata* and *C. krusei* might be an unavoidable result. Lim et al. reported that YS08 caspofungin testing appeared unreliable for *C. glabrata* and that the YS08 micafungin result was more reliable than the caspofungin result (17). Clinical echinocandin resistance is generally associated with amino acid substitutions in specific hot spot regions of *FKS1* (all *Candida* species) and *FKS2* (*C. glabrata* only) (24). CLSI guidelines recommend that if the caspofungin result is susceptible, it can be reported as susceptible, but the possibility of "intermediate" or "resistance" should be confirmed through a micafungin or anidulafungin test or DNA sequence analysis of *FKS* genes (14). Similar to previous research using the EUCAST method, we found that antifungal susceptibility to other echinocandins or an *FKS* genetic study should be performed and interpreted comprehensively when interpreting echinocandin resistance (23, 25).

We performed antifungal susceptibility tests on 45 *C. auris* strains collected in Korea from 2017–2020, and they showed excellent EA except for micafungin (82.2%). A previous study in Korea that used 61 *C. auris* isolates collected from 1996–2018 showed excellent EAs (>96.7%) and CAs (>93.4%) for fluconazole, amphotericin B, caspofungin, and micafungin between the BMD and the previous version of YS07 (26). However, several surveillance reports have revealed consistently high fluconazole MICs and variable resistance to echinocandins and amphotericin B and the echinocandins in *C. auris* (12, 27–30). Moreover, multidrug-resistant strains have emerged in the Americas, Africa, Asia, and Europe (27, 28, 30). Higher MIC50 values of YS07 for amphotericin B, caspofungin, and voriconazole were previously reported compared with our results of 8 $\mu$g/mL, 0.5 $\mu$g/mL, and 1 $\mu$g/mL (12). However, there were no cases of echinocandin, amphotericin B, or multidrug-resistant *C. auris* strains in Korea (26). We observed a lower level of resistance to fluconazole and resistance to amphotericin B at a level of 24.4%. Similar to a previous study (26), our results found that *C. auris* in Korea had relatively low resistance to antifungal agents compared with those isolated from other regions. Multidrug-resistant strains of *C. auris* have been mainly identified among the South American clade (clade IV) or South Asia clade (clade I). We did not identify the clade of *C. auris* in this study, but previous studies have identified that Korean isolates mainly belonged to the East Asia clade (clade II) (26).

As in a previous paper (6), we used ECV/CBP of *C. albicans* to identify rare yeast species. In fact, cutoffs are species-specific, so it may be unwise to apply them to other species arbitrarily. Since the WT of *C. albicans* was universally susceptible to the antifungal agents in this study, we posit that a method to evaluate the aberrant MIC distribution based on *C. albicans* ECV/CBP is a meaningful approach (6).

In conclusion, YS08 showed comparable results with the BMD, including for *C. auris*. Voriconazole testing of YS08 with the method-specific CBP or by applying an ECV of 0.5 $\mu$g/mL may allow for reliable discrimination of fluconazole-resistant *C. glabrata*. However, interpreting the results of this method using the CBPs/ECVs determined for other methods may lead to erroneous results. Considering the lower YS08 fluconazole

MIC results compared with BMD in *Candida* species and the YS08 caspofungin results in *C. glabrata* and *C. krusei*, further improvements for the detection of fluconazole and echinocandin resistance are needed.

## MATERIALS AND METHODS

**Clinical samples.** A total of 200 clinical isolates were submitted to the International St. Mary's Hospital from nine universities/general hospitals between July 2017 and June 2020 through the Korean Nationwide Fungal Collection Network. Among them, only species with more than five independent isolates were included in the current study ($n = 184$). The species consisted of *C. auris* ($n = 45$), *C. albicans* ($n = 24$), *C. tropicalis* ($n = 21$), *C. glabrata* ($n = 20$), *C. parapsilosis* ($n = 19$), *C. krusei* ($n = 15$), *Cyberlindnera fabianii (C. fabianii)* ($n = 12$), *C. guilliermondii* ($n = 9$), *C. pelliculosa* ($n = 8$), *C. lusitaniae* ($n = 6$), and *C. orthopsilosis* ($n = 5$). These isolates were recovered from clinical specimens including the ear ($n = 51$), blood ($n = 45$), urine ($n = 34$), sputum ($n = 21$), abscess ($n = 11$), vaginal swab ($n = 9$), body fluids ($n = 6$), catheters ($n = 4$), and wounds ($n = 3$). Colonies in the primary media were subcultured in Sabouraud Dextrose Agar (SDA) for 24 h at 35°C. The identification of clinical isolates was performed as described in a previous study, using matrix-assisted laser desorption/ionization time-of-flight mass spectrometry systems (ASTA MicroIDSys System, ASTA Inc., Suwon, South Korea) and/or molecular sequencing (31). After identification, the strains were frozen at $-70$°C in a glycerol broth (20%) until they were analyzed. This study was approved by the International St. Mary's Hospital, Catholic Kwandong University College of Medicine in Korea (IS21EISI0040).

**Antifungal susceptibility testing and determination of MICs.** The *in vitro* antifungal susceptibility tests for fluconazole, voriconazole, micafungin, caspofungin, amphotericin B, and flucytosine were performed according to the CLSI M27-ED4 (32) and Vitek 2 system (AST-YS08 card: bioMérieux, France) per the manufacturer's instructions. For the BMD, the drug concentration ranges were 0.06–32 mg/L (amphotericin B), 0.12–64 mg/L (fluconazole), 0.03–8 mg/L (voriconazole), 0.008–8 mg/L (caspofungin, and micafungin), and 0.06–64 mg/L (fluconazole and flucytosine). The AST-YS08 (YS08) card contained serial dilution ranges of antifungal concentrations for amphotericin B, fluconazole, voriconazole, micafungin, caspofungin, and flucytosine (0.25–16 $\mu$g/mL, 0.5–64 $\mu$g/mL, 0.12–8 $\mu$g/mL, 0.06–8 $\mu$g/mL, 0.12–8 $\mu$g/mL, and 1–64 $\mu$g/mL, respectively. *C. parapsilosis* (ATCC 22019) and *C. krusei* (ATCC 6258) were included in each test as control isolates. MICs were determined using the CLSI method, and isolates were classified according to the CBP (14) and ECV (15). Isolates with MICs equal to or less than ECV concentrations were defined as WT isolates, and the rest were considered non-WT isolates (15). As CLSI does not provide CBP for voriconazole on *C. glabrata*, the voriconazole MICs were determined with the method-specific CBP of YS08 (susceptible: $\leq 1$ $\mu$g/mL; intermediate: 2 $\mu$g/mL; resistant: $\geq 4$ $\mu$g/mL) according to the manufacturer's instructions. Because CBPs for *C. lusitaniae*, *C. orthopsilosis*, *C. pelliculosa* and *C. fabianii* and ECVs for *C. pelliculosa* and *C. fabianii* were not available, the value from *C. albicans* was used to identify isolates/species with elevated MICs (6). In the case of *C. auris*, MIC breakpoints recommended by the CDC were used: fluconazole $\geq 32$ $\mu$g/mL; amphotericin B $\geq 2$ $\mu$g/mL; caspofungin $\geq 2$ $\mu$g/mL; micafungin $\geq 4$ $\mu$g/mL; amphotericin B and flucytosine: not available (https://www.cdc.gov/fungal/candida-auris/c-auris-antifungal.html). As there was no CLSI CBP or ECV for flucytosine, we applied an ECV of 1 $\mu$g/mL as we did in a previous study (6). The YS08 does not provide fluconazole results for *C. glabrata*, and there is no fluconazole susceptibility category for *C. glabrata* in revised CLSI M60 (14). We alternately predicted using the method-specific CBP of YS08 voriconazole, the current CLSI M59 ECV (0.25 $\mu$g/mL) of voriconazole, and the previous CLSI M27 ECV (0.5 $\mu$g/mL) of voriconazole. For discrepant results, YS08 and BMD were repeatedly tested, and the second-run results were accepted as the final results.

**Data analysis.** MICs that were high off-scale were converted to the next highest level concentrations, and the low off-scale MICs remained unchanged (33). The EA was defined as the difference between MIC dilutions of less than two. The BMD low off-scale MICs of flucytosine, caspofungin, and micafungin, which were lower than the 2-fold dilution scale of YS08, were rounded up to the next highest 2-log dilution to simplify comparisons. The CA was defined as when the MIC results belonged to the same categories according to the CLSI CBP or/and ECV. Very major errors, major errors, and minor errors were defined with BMD as the reference method. Very major errors were those where the reference method classified an isolate as resistant and YS08 classified it as susceptible. If the reference method classified an isolate as susceptible and YS08 classified it as resistant, it was classified as a major error. Minor errors were defined as when one method classified an isolate as susceptible or resistant and the other assay classified it as intermediate or susceptible depending on the dose (19).

## SUPPLEMENTAL MATERIAL

Supplemental material is available online only.
**SUPPLEMENTAL FILE 1,** PDF file, 0.2 MB.

## ACKNOWLEDGMENTS

This research was supported by the Fungi Specialized Pathogen Resource Bank under National Culture Collection for Pathogens (NCCP) of the Korea Disease Control and Prevention Agency (KDCA) (SPRB-2017-02).

We have no conflicts of interest.

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
