## [Reviewer comments · Microbiology Spectrum]

Microbiology Spectrum

Comparison of six antifungal susceptibilities of 11 *Candida* species using the VITEK2 AST-YS08 card and broth microdilution method

Hyeyoung Lee, Seong Hyouk Choi, Junsang Oh, Jehyun Koo, Hyun Ji Lee, Sung-II Cho, Jeong Hwan SHIN, Hae Kyung LEE, Soo-Young Kim, Chae Lee, Young Ree Kim, Yong-Hak Sohn, Woo Jin Kim, Sook-Won Ryu, Gi-Ho Sung, and Jayoung Kim

Corresponding Author(s): Jayoung Kim, International St. Mary's Hospital, College of Medicine, The Catholic Kwandong University

Review Timeline:

Submission Date:	August 11, 2021
Editorial Decision:	October 22, 2021
Revision Received:	December 16, 2021
Accepted:	March 9, 2022

Editor: Slavena Vylkova

Reviewer(s): The reviewers have opted to remain anonymous.

Transaction Report:

DOI: <https://doi.org/10.1128/spectrum.01253-21>

October 22, 2021

Dr. Jayoung Kim
International St. Mary's Hospital, College of Medicine, The Catholic Kwandong University
Department of Laboratory Medicine
Incheon
Korea (South), Republic of

Re: Spectrum01253-21 (Comparison of six antifungal susceptibilities of 10 *Candida* species as determined by the VITEK2 AST-YS08 card and broth microdilution method)

Dear Dr. Jayoung Kim:

Thank you for submitting your manuscript to Microbiology Spectrum. When submitting the revised version of your paper, please provide (1) point-by-point responses to the issues raised by the reviewers as file type "Response to Reviewers," not in your cover letter, and (2) a PDF file that indicates the changes from the original submission (by highlighting or underlining the changes) as file type "Marked Up Manuscript - For Review Only". Please use this link to submit your revised manuscript - we strongly recommend that you submit your paper within the next 60 days or reach out to me. Detailed information on submitting your revised paper are below.

Link Not Available

Sincerely,

Slavena Vylkova

Journals Department
Reviewer comments:

Reviewer #1 (Comments for the Author):

Comparison of six antifungal susceptibilities of 10 *Candida* species as determined by the VITEK2 AST-YS08 card and broth microdilution method.

The authors present a study comparing two methods of MIC determination for clinical isolates. This kind of study can be useful for clinicians but, from my point of view the results are difficult to understand as many numbers and abbreviations are used. It might be useful to make a graph that is easier to visualise. Also the materials and methods lack technical details. Some statistical analysis should be useful.

check the spelling of *Candida krusei* throughout the text

Candida should be in italic

In the tables, MIC unit (mg/L) are missing

Line 53 : not clear that the BMD is the one standardized by CLSI

Line 69 : I don't really understand the sense of the sentence because it is known that breakpoint of ECV must be used with the method for which they were determined.

MM

In the Material and Methods section, Could you please give more details about the list of species, in the title there is 10 Candida species. In the table 1 there is 10 species but not all Candida and in the text and table 4 there is information about an additional species of Canida.

Line 94 species distribution is also different according to the geographic origin

Line 113 ECV are not really used to determined resistance but to identify isolates having a non-wild type profile

Line 121 broth microdilution standardized EUCAST also exist for the antifungal susceptability testing of yeast.

Line 123 In my opinion, there is a nonsense in the sentence because the MIC is not determined according to the BP

Line 183 Do you mean that CBP classify isolates as resistant or susceptible ?

Line 241 Similar results have been observed using EUCAST method with many interlaboratory variation for MIC caspofungin.

EUCAST AFST propose an ATU (area of technical uncertainty) for anidulafungin isolates determined as susceptible but with MIC micafungin >0.03mg/L, those isolates have FKS mutation and could be identified as resistant.

Line 253 Candida auris multidrug resistant have been mainly identified among the South American clade. Do you know the clade of the Korean isolates recovered in this study ?

Line 282 in the introduction, authors only cite invasive infection but many clinical isolates seem to be involved in colonization.

Line 307 close the bracket after respectively

Correct « *C. krusei* ATCC (ATCC 6258) » by « *C. krusei* (ATCC 6258) »

Line 308 MICs are not determined by CLSI breakpoint, MICs are determined using CLSI method and isolates are classified according to the BP.

Line 315 about *C. pelliculosa* and *C. fabianii* : knowin that these species don't belong to the same clade than *C. albicans* (different genus) I am not sure that the use of *C. albicans* BP is very reliable. Maybe the calculation of MIC90 or local ECOFF should be more apporpritate to determine isolates with high MIC

Were the MICs for isolates with discordant results tested from isolated colonies, are you sure that the original isolates (especially those recovered from a non-sterile site) were pure, as if you have a mixture of species or sub-populations, the MICs may be different.

Reviewer #2 (Comments for the Author):

The article by Lee etal provides an interesting comparison of two well established laboratory techniques used in characterizing the sceptibilities of Candida clinical isolates. The authors showed the comparisons between Vtek2 and Broth microdilution method for six well known antifungals. Here are my comments:

1)The introduction needs some work. The introduction will greatly benefit if the authors briefly describe the different antifungals that used in treating fungal infections and briefly describe the common resistance mechanisms to each drug as described previously in Candida

<https://pubmed.ncbi.nlm.nih.gov/32441527/>

<https://pubmed.ncbi.nlm.nih.gov/32526921/>

2)Line 99: " Antifungal resistance is rare in *C. albicans*".. While echinocandin resistance is rare in *C. albicans*, fluconazole resistance is quite common. The authors should reconsider the statement.

3)Though the authors have mentioned the breakpoints for individual antifungal drugs in the material and methods for individual organisms, a table providing the cutoffs for each drug/organism will greatly improve the clarity of the manuscript.

4)From the introduction the rationale of the study is not clear. By comparing the two methods does the author hypothesize that Vitek2 can be used in place of BMD to analyze susceptibility profiles in different Candida sp? or does the author hypothesize that clinicians should consider using both the techniques simultaneously? please clarify

5)All organism names in Table 1 need to be italicized.

6)Line 226: *C. krusei* is mistyped.

7)Lines 226-228: Caspofungin can show "eagle effects" , which may cause variability in MICs as described previously (<https://pubmed.ncbi.nlm.nih.gov/16569843/>) This needs to be discussed for better clarity and will make the manuscript stronger.

8)Lines 237-241: Why the possibility of intermediate resistance should be confirmed with micafungin or anidulafungin test?

Please clarify

9)Line 239-241: "When interpreting... interpreted comprehensively" Is there any previous evidences for this statement? If so please clarify and provide references.

Staff Comments:

Preparing Revision Guidelines

Please return the manuscript within 60 days; if you cannot complete the modification within this time period, please contact me. If you do not wish to modify the manuscript and prefer to submit it to another journal, please notify me of your decision immediately so that the manuscript may be formally withdrawn from consideration by Microbiology Spectrum.

Response to the reviewer

Reviewer 1

The authors present a study comparing two methods of MIC determination for clinical isolates. This kind of study can be useful for clinicians but, from my point of view the results are difficult to understand as many numbers and abbreviations are used. It might be useful to make a graph that is easier to visualise. Also the materials and methods lack technical details. Some statistical analysis should be useful.

-> Thank you for your comment. We understand your point and have considered changing some tables into figures based on your comments. However, it was not easy because some information was omitted and missing information occurred. Instead, the text content was systematically revised and corrected to make it easier to understand. We also modified Table 2 to make it easier to understand. We apologize for not being able to correct the reviewer's comments.

In the tables, MIC unit (mg/L) are missing

-> Thank you for your comment. According to the reviewer's comment, we added the MIC unit to the tables.

Line 53: not clear that the BMD is the one standardized by CLSI

-> Thank you for your comment. According to the reviewer's comment, we corrected the sentence as "We used a VITEK 2 AST-YS08 (YS08) system and the broth microdilution method (BMD) adopted by the Clinical and Laboratory Standards Institute (CLSI)" in lines 51-52.

Line 69: I don't really understand the sense of the sentence because it is known that breakpoint of ECV must be used with the method for which they were determined.

-> Thank you for your comment. According to the reviewer's comment, we corrected the sentence as "YS08 showed comparable results to the BMD. However, considering the lower

YS08 fluconazole MIC results compared with BMD in *Candida* species and YS08 caspofungin results in *C. glabrata* and *C. krusei*, improvements are needed.” in lines 67-70.

In the Material and Methods section, Could you please give more details about the list of species, in the title there is 10 *Candida* species. In the table 1 there is 10 species but not all *Candida* and in the text and table 4 there is information about an additional species of *Canida*.

-> According to the reviewer`s comment, we added the full list of species in line 295-298 as “The species consisted of *C. auris* (n=45), *C. albicans* (n=24), *C. tropicalis* (n=21), *C. glabrata* (n=20), *C. parapsilosis* (n=19), *C. krusei* (n=15), *Cyberlindnera fabianii* (*C. fabianii*) (n=12), *C. guilliermondii* (n=9), *C. pelliculosa* (n=8), *C. lusitaniae* (n=6), and *C. orthopsilosis* (n=5).

-> We changed the title to “11 *Candida* species”

Line 94 species distribution is also different according to the geographic origin

-> According to the reviewer`s comment, we corrected the sentence as “*Candida albicans* was previously the predominant pathogen (3, 5), but in recent years, *C. glabrata*, *C. tropicalis*, *C. krusei*, *C. parapsilosis*, and *C. lusitaniae* have emerged as important pathogens (6).” in line 99-101.

Line 113 ECV are not really used to determined resistance but to identify isolates having a non-wild type profile

-> According to the reviewer`s comment, we corrected the sentence as “The epidemiological cutoff value (ECV) identifies isolates that have a non-wild type (WT) profile,” in line 127.

Line 121 broth microdilution standardized EUCAST also exist for the antifungal susceptibility testing of yeast.

-> According to the reviewer`s comment, we corrected the sentence as “The aim of this study was to evaluate the clinical applicability of the new VITEK 2 AST-YS08 (YS08) card by

comparing it with the results of the BMD method by CLSI.” in line 135-136.

Line 123 In my opinion, there is a nonsense in the sentence because the MIC is not determined according to the BP

-> According to the reviewer`s comment, we corrected it as “MIC distributions for each *Candida* species were determined by CLSI reference broth microdilution method.” in line 139-140.

Line 183 Do you mean that CBP classify isolates as resistant or susceptible ?

-> According to the reviewer`s comment, we corrected as “The CBPs were used to define susceptible or resistant isolates.” in line 196.

Line 241 Similar results have been observed using EUCAST method with many interlaboratory variation for MIC caspofungin. EUCAST AFST propose an ATU (area of technical uncertainty) for anidulafungin isolates determined as susceptible but with MIC micafungin >0.03mg/L, those isolates have FKS mutation and could be identified as resistant.

-> According to the reviewer`s comment, we corrected as “Similar to previous research using the EUCAST method, we found that antifungal susceptibility to other echinocandins or an FKS genetic study should be performed and interpreted comprehensively when interpreting echinocandin resistance (23, 25).” in line 254-257.

Line 253 *Candida auris* multidrug resistant have been mainly identified among the South American clade. Do you know the clade of the Korean isolates recovered in this study ?

-> Thank you for your precise comment. Unfortunately, we did not identify the clade of *C. auris* in this study. Previous studies identified that Korean isolates belonged to clade II and all isolates from Korean hospitals had quite different EK and REAG-N patterns from CDC *C. auris* isolates of the other three clades (clades I, III, and IV). (J Clin Microbiol. 2019 Mar 28;57(4):e01624-18)

-> We added your comment in the discussion as “Multidrug-resistant strains of *C. auris* have been mainly identified among the South American clade (clade IV) or South Asia clade (clade I). We did not identify the clade of *C. auris* in this study, but previous studies have identified that Korean isolates mainly belonged to the East Asia clade (clade II) (26).” in lines 272-275.

Line 282 in the introduction, authors only cite invasive infection but many clinical isolates seem to be involved in colonization.

-> Thank you for your precise comment. We added in lines 93-95 as “*Candida* species are normal commensals that localize on the skin and mucosal membranes of genitals and the gastrointestinal tract. However, they can cause various infections in vulnerable patients, such as the elderly, hospitalized, or immunosuppressed”.

Line 307 close the bracket after respectively

Correct « *C. krusei* ATCC (ATCC 6258) » by « *C. krusei* (ATCC 6258) »

-> Thank you for your comment. We corrected it in line 319.

Line 308 MICs are not determined by CLSI breakpoint, MICs are determined using CLSI method and isolates are classified according to the BP.

-> Thank you for your comment. According to the reviewer`s comment, we corrected it as “MICs were determined using the CLSI method, and isolates were classified according to the CBP (14) and ECV (15).” in line 319-321.

Line 315 about *C. pelliculosa* and *C. fabianii* : knowin that these species don't belong to the same clade than *C. albicans* (different genus) I am not sure that the use of *C. albicans* BP is very reliable. Maybe the calculation of MIC90 or local ECOFF should be more appropriate to determine isolates with high MIC

-> Thank you for your comment. According to the reviewer`s comment, we added the limitation of this study as “As in a previous paper (6), we used ECV/CBP of *C. albicans* to

identify rare yeast species. In fact, cutoffs are species-specific, so it may be unwise to apply them to other species arbitrarily. Since the WT of *C. albicans* was universally susceptible to the antifungal agents in this study, we posit that a method to evaluate the aberrant MIC distribution based on *C. albicans* ECV/CBP is a meaningful approach (6).” in lines 276-280.

Were the MICs for isolates with discordant results tested from isolated colonies, are you sure that the original isolates (especially those recovered from a non-sterile site) were pure, as if you have a mixture of species or sub-populations, the MICs may be different.

-> In case of discrepancy, we performed retests using the same strain solution at the same time. Since this study is to find the difference between the two test methods, the discrepancy does not appear to be due to drug resistance subpopulations in heterogenous single-species infection.

Reviewer #2 (Comments for the Author):

1) The introduction needs some work. The introduction will greatly benefit if the authors briefly describe the different antifungals that used in treating fungal infections and briefly describe the common resistance mechanisms to each drug as described previously in *Candida*

<https://pubmed.ncbi.nlm.nih.gov/32441527/>

<https://pubmed.ncbi.nlm.nih.gov/32526921/>

-> Thank you for your comment. We describe the different antifungal agents and the common resistance mechanisms in Introduction as “Antifungal agents target various biosynthetic pathways of pathogens. Echinocandins target biosynthesis of the cell wall. Azoles target the important enzyme, 14 α -demethylase, in ergosterol biosynthesis. Flucytosine (5-FC) interferes with nucleic acid biosynthesis. Polyene drugs, including amphotericin B, bind with ergosterol to form pores and are fungicidal (1, 9).” in lines 107-110.

“The molecular mechanisms involved in antifungal resistance include overexpression of membrane transporters, changes in the biosynthesis of the cell wall and ergosterol, mutations in the transcription factors that regulate membrane transporters, and ergosterol biosynthesis (9). in line 114-117.

2) Line 99: " Antifungal resistance is rare in *C. albicans*". While echinocandin resistance is rare in *C. albicans*, fluconazole resistance is quite common. The authors should reconsider the statement.

-> Thank you for your comment. We deleted that sentence and added a new sentence as "An important factor that may contribute to therapeutic failure is antifungal agent resistance (1)." in line 111.

3) Though the authors have mentioned the breakpoints for individual antifungal drugs in the material and methods for individual organisms, a table providing the cutoffs for each drug/organism will greatly improve the clarity of the manuscript.

-> Thank you for your comment. According to the reviewer's comment, we provided the cutoffs for each drug/organism in table 1 and table 3.

4) From the introduction the rationale of the study is not clear. By comparing the two methods does the author hypothesize that Vitek2 can be used in place of BMD to analyze susceptibility profiles in different *Candida* sp? or does the author hypothesize that clinicians should consider using both the techniques simultaneously? please clarify

-> Thank you for your comment. According to the reviewer's comment, we modified the purpose of the study as "The aim of this study was to evaluate the clinical applicability of the new VITEK 2 AST-YS08 (YS08) card by comparing it with the results of the BMD method by CLSI." in line 135-136.

And we concluded that "In conclusion, YS08 showed comparable results with the BMD, including for *C. auris*. Voriconazole testing of YS08 with the method-specific CBP or by applying an ECV of 0.5 µg/ml may allow for reliable discrimination of fluconazole-resistant *C. glabrata*. However, interpreting the results of this method using the CBPs/ECVs determined for other methods may lead to erroneous results. Considering the lower YS08 fluconazole MIC results compared with BMD in *Candida* species and the YS08 caspofungin results in *C. glabrata* and *C. krusei*, further improvements for the detection of fluconazole and

echinocandin resistance are needed.” in line 281-287.

5) All organism names in Table 1 need to be italicized.

-> Thank you for your comment. We corrected the name of all organism to be italicized.

6) Line 226: *C. krusei* is mistyped.

-> Thank you for your comment. We corrected the typo.

7) Lines 226-228: Caspofungin can show "eagle effects" , which may cause variability in MICs as described previously (<https://pubmed.ncbi.nlm.nih.gov/16569843/>) This needs to be discussed for better clarity and will make the manuscript stronger.

-> Thank you for your comment. According to the reviewer`s comment, we discussed the eagle effects in caspofungin as “Caspofungin can show "eagle effects," which may cause variability in MICs, as described previously (21). When susceptibility tests are performed under standardized conditions, a surprising echinocandin-specific paradoxical effect can be observed. This effect refers to a phenomenon in which certain strains grow in higher concentrations of echinocandin while being completely susceptible at lower concentrations. This has been observed in several species of *Candida* and *Aspergillus* (22).” in line 236-241.

8) Lines 237-241: Why the possibility of intermediate resistance should be confirmed with micafungin or anidulafungin test? Please clarify

-> Thank you for your comment. We clarified as “CLSI guideline recommends that if the caspofungin result is susceptible, it can be reported as susceptible, but the possibility of “intermediate” or “resistance” should be confirmed through a micafungin or anidulafungin test or DNA sequence analysis of FKS genes (14).” in line 251-254.

9) Line 239-241: "When interpreting... interpreted comprehensively" Is there any previous evidences for this statement? If so please clarify and provide references.

-> Thank you for your comment. We modified the sentence as “Similar to previous research using the EUCAST method, we found that antifungal susceptibility to other echinocandins or an FKS genetic study should be performed and interpreted comprehensively when interpreting echinocandin resistance (23, 25).” in line 254-257.

March 9, 2022

Dr. Jayoung Kim
International St. Mary's Hospital, College of Medicine, The Catholic Kwandong University
Department of Laboratory Medicine
Incheon
Korea (South), Republic of

Re: Spectrum01253-21R1 (Comparison of six antifungal susceptibilities of 11 *Candida* species using the VITEK2 AST-YS08 card and broth microdilution method)

Dear Dr. Jayoung Kim:

Your manuscript has been accepted, and I am forwarding it to the ASM Journals Department for publication. You will be notified when your proofs are ready to be viewed.

Sincerely,

Slavena Vylkova
Editor, Microbiology Spectrum

Journals Department
Supplemental file 1: Accept